# Enhanced Performance of Chitosan via a Novel Quaternary Magnetic Nanocomposite Chitosan/Grafted Halloysitenanotubes@ZnγFe_3_O_4_ for Uptake of Cr (III), Fe (III), and Mn (II) from Wastewater

**DOI:** 10.3390/polym13162714

**Published:** 2021-08-13

**Authors:** Mahmoud F. Mubarak, Ahmed H. Ragab, Rasha Hosny, Inas A. Ahmed, Hanan A. Ahmed, Salah M. El-Bahy, Abeer El Shahawy

**Affiliations:** 1Petroleum Application Department, Egyptian Petroleum Research Institute (EPRI), Nasr City, Cairo 11727, Egypt; 2Department of Chemistry, Faculty of Science, King Khalid University, Abha 62224, Saudi Arabia; ahrejab@kku.edu.sa (A.H.R.); eaahmed@kku.edu.sa (I.A.A.); 3Production Department, Egyptian Petroleum Research Institute (EPRI), Nasr City, Cairo 11727, Egypt; rasha@epri.eg.org; 4Petrochemicals Department, Egyptian Petroleum Research Institute (EPRI), Nasr City, Cairo 11727, Egypt; Hanan@epri.eg.org; 5Department of Chemistry, Turabah University College, Taif University, P.O. Box 11099, Taif 21944, Saudi Arabia; s.elbahy@tu.edu.sa; 6Department of Civil Engineering, Faculty of Engineering, Suez Canal University, Ismailia 41522, Egypt

**Keywords:** chitosan, halloysite, quaternary magnetic nanocomposite, heavy metals, wastewater treatment

## Abstract

A novel chitosan/grafted halloysitenanotubes@Znγmagnetite quaternary nanocomposite (Ch/g-HNTs@ZnγM) was fabricated using the chemical co-precipitation method to remove the ions of Cr (III), Fe (III), and Mn (II) from wastewater. The characteristics of the synthesized Ch/g-HNTs@ZnγM quaternary nanocomposite were investigated using FTIR, SEM, XRD, GPC, TGA, TEM, and surface zeta potential. The characterization analysis proved that the mentioned nanocomposite structure contains multiple functional groups with variable efficiencies. Additionally, they proved the existence of magnetic iron in the nanocomposite internal structure with the clarity of presentation of gaps and holes of high electron density on its surface. The results showed that the pH and time to reach an equilibrium system for all the studied metal ions were obtained at 9.0 and 60 min, respectively. The synthesized Ch/g-HNTs@ZnγM nanocomposite exhibited maximum adsorption removal of 95.2%, 99.06%, and 87.1% for Cr (III), Fe (III), and Mn (II) ions, respectively. The pseudo-second-order kinetic model and, for isotherm, the Langmuir model were best fitted with the experimental data. The thermodynamic parameters indicated the exothermic and spontaneous nature of the adsorption reaction as proven by the ΔH° and ΔG° values. Additionally, chemical adsorption by the coordination bond is supposed as the main mechanism of adsorption of the mentioned metal ions on the nanocomposite. Finally, Ch/g-HNTs@ZnγM displays prospected advantages, such as a low-expense adsorbent, high efficiency and availability, and an eco-friendly source, that will reduce the environmental load via an environmentally friendly method.

## 1. Introduction

Among the various sources of water pollution with heavy metals, the rapid increase in industrialization is causing many environmental problems day by day [1]. Water pollution has reached a point of threatening human and aquatic life. Heavy metals accumulate in living organisms because of their toxicity and non-biodegradability [2,3]. Therefore, they can cause a lot of acute and chronic diseases. The treatment of industrial wastewater including heavy metals is becoming an important subject for improving water quality. Cd (II), Cr (III), Cu (II), Fe (III), Mn (II), and Ni (II) are considered as some of the most toxic metals for living organisms when their concentration exceeds the required limits [4]. Increased chromium exposure leads to cancer, asthma, and diarrhea. Physiological deficiency is caused by liver disease, kidney complications, and a brain defect [5,6]. The presence of iron in wastewater above the tolerable level can cause intestinal damage and respiratory tract irritation [7]. Additionally, the contamination of water with Mn (II)/Fe (III) ions leads to poor water taste, odour, colour, and turbidity; Mn (II)/Fe (III) water pollution also causes many chronic diseases [8].

Chemical precipitation, flotation, reverse osmosis, ultrafiltration, electrochemical precipitation, ion exchange, chemical oxidation, adsorption, and other conventional treatment methods are generally used to remove heavy metals from wastewater [9,10]. Most of the mentioned techniques have ingrained restrictions, such as low efficiency, the generation of significant quantities of sludge, and costly disposal, except adsorption [9]. Adsorption is the most preferred among the mentioned treatment technologies for extracting heavy metals from wastewater because of its efficiency [10]. It provides design flexibility, and the treated effluents are of high quality, are reversible, and absorbent materials can be regenerated [11,12]. Among the substances used to remove metal ions from contaminated water by adsorption are alumina, silica, activated carbon, graphene oxide, manganese oxides, ferric oxide nanoparticles, polyaniline, titanium oxides, and zinc oxide [13,14,15]. Lately, scientists have focused on preparing new multifunctional adsorbents with low costs and ease of use [16,17]. These adsorbents may be composed of synthetic or natural substrates, such as industrial byproducts, clays, biosorbents, and modified biosorbents [18,19]. Among these substrates, chitosan (natural biopolymers) [20] is considered an effective adsorbent for removing heavy metals [21] because it is a polysaccharide with the functional groups -OH and -NH_2_, is hydrophilic and biodegradable, and is easy to derive because of its ability to form chelates with the heavy metals [22,23]. However, it is known for its mechanical weakness, [23,24] its ability to dissolve in an acidic medium, and its capability of leaching carbohydrates if utilized in its raw structure [25]. Therefore, several adsorbents, including “magnetic” nanoparticles and chitosan, have been prepared and have shown their excellent efficiency for heavy metal removal from wastewater [26,27]. Magnetic adsorbents are considered alternative adsorbents because of their fascinating properties, i.e., high adsorption performance and magnetic characteristics, facilitating their separation by applying an external magnet [28,29]. Magnetic gamma iron oxide (γ ferric oxide Fe_2_O_3_) is one of the iron oxides which has a high magnetic field compared to the other three iron oxides, i.e., Fe_3_O_4_ (iron (II, III) oxide) and FeO (iron (II) oxide) [30,31]. When γ Fe_2_O_3_ (ferric oxide) is embedded with natural polymers, it gives a high potential nanocomposite that can be used in many fields such as medicine and water treatment due to its high ability to adsorb organic and inorganic pollutants [32].

Based on the above-mentioned concerns, this paper focused on searching for low cost-effective adsorbents and eco-friendly methods to treat wastewater from inorganic effluent (heavy metals) [33]. Herein, chitosan extract was abstracted for shrimps [34]. A novel adsorbent quaternary nanocomposite, chitosan/grafted halloysitenanotubes@Znγmagnetite (Ch/g-HNTs@ZnM), was synthesized via the ultrasonic-assisted adhesion technique to adsorb the Cr (III), Fe (III), and Mn (II) from wastewater. The characteristics of chitin (chitosan extract), halloysite, Zn@Fe_3_O_4_, and the synthesized Ch/g-HNTs@ZnγM quaternary nanocomposite were revealed by several techniques, including FTIR, SEM, XRD, GPC, TGA, TEM, and surface zeta potential. Batch processes with the synthesized Ch/g-HNTs@ZnγM were investigated. Generally, adsorption via the continuous column process is the most promising method for the future wastewater treatment implementations of existing adsorbents. The various factors affecting the metal ion removal efficacy on the Ch/g-HNTs@ZnγM adsorbent were studied. The effects of solution pH, metal ions’ initial concentration, temperature, quaternary adsorbent dosage, and contact time were investigated for batch adsorption. Additionally, nonlinear regression was designed for various kinetic and equilibrium models to evaluate the adsorption data.

## 2. Experiments

### 2.1. Materials

The authors have utilized sodium hydroxide (NaOH) (99.99%, Sigma Aldrich, Munich, Germany), hydrochloric acid (HCl), calcium carbonate (CaCO_3_), potassium hydroxide (KOH), distilled water, deionized (DI) water, acetic acid, an aqueous solution of sodium dodecyl sulfate (SDS), Ethylenediaminetetraacetic acid (EDTA), ferric nitrate Fe(NO_3_)_3_, zinc nitrate Zn(NO_3_)_2_, and halloysite solution to complete the following experiments.

### 2.2. Extraction of Chitosan

Fresh shrimp shells were locally obtained from an Egyptian market in Cairo. After removing their legs and heads, the shells were well rinsed repeatedly with tap water, followed by their drying at 70 °C and grinding into fine powder [35].

Chitosan (Ch) was then extracted from the shrimp shell powder according to Hosny et al., 2014 [20]. First, the deproteinization process was performed by adding 0.1 N NaOH solution to the shrimp shells with continuous stirring for two days to remove proteins, filtration of the produced solution at 25 °C, and washing the filter with distilled water to the pH of 7. Second, the demineralization process was carried out by stirring the resulting filter for 8 continuous days in 200 mL of a 3–5% hydrochloric acid solution, then filtering and washing to eliminate the minerals and obtain pure chitin. Finally, the deacetylation was achieved by hydrolyzing the produced chitin in 50% NaOH and stirring for 4 h at a fixed temperature of 40 or 90 °C, followed by washing the filter with water at 50 °C while stirring for 5 h to produce pure chitosan at a pH of 7. The mentioned steps for pure chitosan extraction from shrimp shells are depicted in Figure 1.

### 2.3. Synthesis of Halloysite Nanotubes/ZnγFe_3_O_4_ Core/Shell (HNPs/ZnγM)

Halloysite suspension solution was prepared by vigorously stirring 5.0 g halloysite in a 200 mL aqueous solution. Then, the halloysite solution was impregnated by ZnγFe_3_O_4_ particles to produce (HNTs/ZnγM) core/shell using the chemical precipitation method. In detail, zinc/iron precursors solution consisting of ferric nitrate (Fe(NO_3_)_3_ and zinc nitrate (Zn(NO_3_)_2_ was added dropwise to the 1% halloysite suspension solution. The (Fe(NO_3_)_3_:(Zn(NO_3_)_2_) stoichiometric ratio was 4:1 in the solution. The final black (HNTs/ZnγM) suspension was rinsed repeatedly with re-distilled water, dried at 50 °C for 24 h and at different temperatures (100 °C to ~500 °C, the interval was 50 °C), and calcined for 2 h in the air with consideration of a 1 °C/min heating rate.

### 2.4. Synthesis of a Novel Chitosan/Grafted Halloysite@ZnγFe_3_O_4_ Tetra-Nanocomposite (Ch/g-HA@ZnγM)

The ultrasonic-assisted adhesion method was used to synthesize the Chitosan/grafted Halloysite@ZnγFe_3_O_4_ tetra-nanocomposite (Ch/g-HNTs@ZnγM) because it diminishes the time of crystallization and forms more consistent nuclei. Briefly, in the first step, the ultrasonic dispersion method was applied for 30 min at 25 °C on a mixture solution containing 0.5 g Fe_3_O_4_-NH_2_ and 0.862 g (3 mmoL) Zn(NO_3_)_2_·6H_2_O, and dissolved in 30 mL of ethanol and containing Zn(NO_3_)_2_·6H_2_O. 0.862 g (3 mmoL). In this step, the magnetic Fe_3_O_4_ in its nanoscale was added to adhere to the surfaces of Metal–Organic Frameworks (MOFs) precursor solution (Zn(NO_3_)_2_·6H_2_O) to form MMOFs (ZnγFe_3_O_4_). Secondly, the same technique was applied on another mixture containing 1.50 g (9 mmoL) chitosan and 0.1 g PVP soluble in 30 mL of ethanol. Then, the mixture solution of the second step was stepwise added to the first one and dispersed together by ultrasonic for 1 h at 25 °C, then aged at 60 °C for 3 h. Lastly, the chitosan/grafted halloysitenanotubes@ZnγFe_3_O_4_ tetra-nanocomposite was produced by washing the mixture solution with anhydrous ethanol and deionized water drying by a vacuum pump at 50 °C for 24 h. The expected schematic diagram of Ch/g-HNTs@ZnγM synthesis is shown in Figure 2 and Figure 3.

### 2.5. Characterization of Materials

The synthesized materials were analyzed with the ATR FTIR spectra (SHIMADZU, IRAffinity-1S, Surgut, Russia) over the wavenumber 4000–400 cm^−1^. The prepared chitosan molecular weight and thermogravimetric analysis (TGA) of chitosan and Ch/g-HNTs@ZnγM nanocomposite were determined by the GPC analysis. A supremamax 3000 column (Mainz, Germany) was used. At the same time, the TGA-50 Shimadzu instrument was under air at a heating rate of 10 °C min^−1^ from 25 °C to 600 °C. The synthesized materials’ surface morphologies were investigated at 120 kV by a Scanning Electron Microscope (SEM, Quanta 450 FEG, FEI Company, North Brabant, Netherlands). XRD diffraction analysis was performed with PAN alytical–Empyrean using Cu Kα radiation (λ = 0.15406 nm, 40 mA, 45 kV, step size (°2Th.) 0.0260). TEM with Model JEM-200CX (JEOL 2100, JEOL, Tokyo, Japan), operated at 200 kV, was used to investigate the morphology of the halloysite and magnetic halloysite samples. An important parameter is the pHpzc of the adsorbents characterized by the Zeta sizer Nano (Malvern, UK), which defines the pH at which the adsorbent surface has net electrical neutrality. The wet Ch/g-HNTs@ZnγM beads surface charge was examined using the method described elsewhere to determine the point of zero charges for the adsorbent. Usually, 0.1 g of the wet synthesized composite was mixed with 40 mL of H_2_O. Each solution’s pH was changed from 3 to 12 using (0.1 M) H_2_SO_4_ or NaOH. Then, the suspensions were shaken at 250 rpm for 48 h before pH equilibrium was reached. The pHpzc was the point where the plot of pH (pH = pH_f_pH_i_) (*Y*-axis) vs. pH_i_ intersected (*X*-axis).

### 2.6. Adsorption Experiments

The batch adsorption studies were carried out in beakers with 25 mL of the studied heavy metals solutions of a certain concentration (15 mg/L) with a specific Ch/g-HNTs@ZnγM adsorbent dosage. Shaking of the solution was performed at a particular rate of 50 rpm for various contact times (10–260 min). Processing factors including initial metals concentrations, PH, temperatures, Ch/g-HNTs@ZnγM tetra-nanocomposite dosage, and time intervals were explored and optimized for studying the maximum heavy metal removal. At the end of shaking, the absorbance values of the studied heavy metals residual were measured at a wavelength of 250–664 nm by a UV-Vis spectrophotometer (JENWAY 6715, Shimadzu, Columbia, MD, USA). Each experiment was repeated in triplicate, and the average values were utilized for analyzing the adsorption data. Various adsorption temperatures in the range of 25–55 °C were examined at a constant Ch/g-HNTs@ZnγM tetra-nanocomposite dosage and heavy metals concentrations to study the adsorption thermodynamics. The studied heavy metal removal percentage (*R*%) was determined by Equation (1) [36].

(1)
$$R\;\%=\frac{\left(C_{0}-C_{e}\right)}{C_{o}}\times 100$$


*C*_o _is the metal ions’ initial concentrations (mg/L), and *C*_e_ is the remaining metal ions concentrations after treatment (mg/L).

## 3. Results and Discussion

### 3.1. Characterization of Materials

#### 3.1.1. Physical and Chemical Characterization of Prepared Chitin and Chitosan

Chitin and chitosan are bio-macromolecules (natural polymers). Their various characteristics, i.e., the prepared chitin and chitosan, are estimated based on FTIR spectra, X-ray diffraction analysis, gel permeation chromatography (GPC), thermal analysis, and a scanning electron microscope (SEM).

#### 3.1.2. Fourier Transform Infrared (FTIR) Spectra

The prepared chitin chemical bonds are investigated by FTIR spectra, as depicted in Figure 4. This spectrum shows a band at 1659 cm^−1^ and an intensive band at 1319 cm^−1^, respectively, correlated to the stretching vibration of CN superimposed to the C=O group linked by a hydrogen bond with the –OH group, CH_3_ group asymmetrical deformation or rocking. The non-existence of a band at 1540 cm^−1^ demonstrated that the successive chitin treatment was strong enough to eliminate all the proteins; thereby, pure chitin was obtained.

Figure 4 also represents the FTIR spectrum of chitosan. In general, chitosan shows bands at 1400–1650 cm^−1^ (C=O bonds), as illustrated by Chatterjee S. et al., 1994 [37]. The broad FTIR bands that appeared in the range of 3000–3500 cm^−1^ are correlated to the stretching vibration of O–H. The characteristic absorption bands of chitosan which appeared at 1319.82 cm^−1^, 2924.40 cm^−1^, and at 1555.69 cm^−1^ and 3263.72 cm^−1^ are correlated to C-H and amino vibrations groups. This spectrum also exhibits two bands at 873.93 cm^−1^, and 1157.57 cm^−1^ that characterize chitosan’s saccharide and polysaccharide structure.

For the halloysite sample (Figure 4), its spectrum emphasizes FTIR bands at 3709 cm^−1^ and 3629 cm^−1^, respectively, due to the inner surface -OH groups and the inner –OH groups stretching vibrations.

For the Ch/g-HNTs@ZnγM magnetic tetra-nanocomposite sample (Figure 4), the appearance of a new band at 532 cm^−1^ reports the Fe–O stretching mode, indicating the existence of Fe_3_O_4_ functional groups. The FTIR band recorded at 1168.71 cm^−1^ was moved to 1630 cm^−1^ with low intensity compared to the chitosan spectrum. The nanocomposite spectrum, the spectrum of the other samples, and the formation of Ch/g-HNTs@ZnγM magnetic tetra-nanocomposites through the N=C groups and the -OH links with halloysite were verified. The Fe-O absorption band of Fe_3_O_4_ appeared at a lower wavenumber than the ZnγFe3O4 sample., possibly because of the electrostatic strength between the surface and the halloysite of ZnγFe_3_O_4_. A Fe-O absorption band was still in the magnetic halloysite chitosan, which suggested using an electrostatic activity to cover the ZnγFe_3_O_4_ particles of halloysite in the magnetic composite.

#### 3.1.3. X-Ray Diffraction (XRD)

The X-ray diffraction (XRD) patterns for chitosan, halloysite, and Zn@Fe_3_O_4_ material provide clear insights into their structural characteristics. The XRD pattern of chitosan shows distinct peaks at 2θ = 9.24° and 17.5°, which correspond to the (010) and (020) diffraction planes. These peaks confirm the semi-crystalline nature of chitosan, with broadening indicating the transformation from the crystalline structure of chitin to the more amorphous structure of chitosan. For halloysite, characteristic reflections at 2θ values of 11.08°, 21.42°, 27.45°, and 32.47° highlight its tubular structure and the crystallinity of this natural nanomaterial. Zn@Fe_3_O_4_ nanoparticles exhibit broad peaks at 2θ = 24.14°, 31.86°, 42.50°, 56.3°, and 61.787°, which are associated with the (210), (220), (400), (511), and (440) planes, confirming the formation of the ZnγFe_3_O_4_ phase, where zinc is doped into the magnetite (Fe_3_O_4_) structure. This confirms the formation of a well-integrated composite material with strong interactions between its components [38,39,40].

#### 3.1.4. Thermal Analysis

The thermal decomposition of chitosan (TGA curve) is shown in Figure 6. In this curve, there are two decomposition stages. The first one begins at 60 °C, with a 6% weight loss due to the water loss. The second one nearly starts at 150 °C with a weight loss of about 47%. It stabilizes at 350 °C, and eventually reaches a total weight loss of 59% at about 500 °C, due to the decomposition of chitosan.

Figure 6 also shows the TGA curve of the magnetic tetra-nanocomposite where the adsorbed water at the surface induces a weight loss at 160 °C. A substantial loss of weight can also occur between 150 °C and 650 °C. A loss of mass by decomposition occurs in the TGA curve, which contains the halloysite and ZnγFe_3_O_4_, caused by the combustion of these fillers gradually to 650 °C.

#### 3.1.5. Gel Permeation Chromatography (GPC)

The average molecular weights of chitosan are determined by the gel permeation chromatography (GPC) using water after 60 min and 120 min, as represented in Table 1 and Table 2. The weight-average molecular weights (Mw) were 86,380 and 110,534, while the number-average molecular weights (Mn) were 50,015 and 51,518 at 60 min and 120 min, respectively.

#### 3.1.6. Scanning Electron Microscope (SEM)

SEM was used to characterize the surface morphology of chitin and chitosan. Figure 7 shows the SEM images of chitosan and chitin. Chitin powder exhibits almost a smooth surface (Figure 7A), while the chitosan powder exhibits a rough surface (Figure 7B). It can be observed that chitosan has an irregular diameter in the range of 14.6–37.6 µm with cracks and pores on its surface, and the number of pores increased significantly, as illustrated in Hosny, 2014 [41]. Consequently, it can be concluded that chitosan has pores and cracks in its surface more than chitin; this gives chitosan a high ability to adsorb heavy metals more than chitin.

SEM images of the magnetic composite were evaluated to describe the size and morphology of these synthesized constructions. According to exact research, the SEM histogram of the magnetic composite is shown in Figure 7. Following the accuracy of the findings, we can see that the magnetic composite was distinguished by a uniform, virtually similar morphology with sphere shapes. It can be noticed that the synthetic nanocomposite filler ZnγFe_3_O_4_/halloysite has a core–shell structure. The average dimensions of the nanocomposite ZnγFe_3_O_4_/halloysite were calculated in a magnetic composite body. The dimensions were between 45 and 62 nm.

#### 3.1.7. TEM

Transmission electron microscopy (TEM) was used for the morphological study of loaded and discharged ZnγFe_3_O_4_ halloysite tubes. The TEM image shows halloysite nanotubes as elongated, cylindrical rods, with diameters ranging between 40 and 80 nm and lengths varying from 500 to 1000 nm (Figure 8). The nanotubes are surrounded by clusters of smaller particles, possibly indicating surface interactions or modifications. The TEM images of the loaded halloysite ZnFe_3_O_4_ have a morphology, validated by TEM analysis, that is random and free of any defects (i.e., beads, globules, and undefined form). It can be concluded that the ZnγFe_3_O_4_-loaded halloysite has similar morphological characteristics (Figure 8). ZnγFe_3_O_4_ was filled with average diameters of 150 ± 45 nm.

#### 3.1.8. Ch/g-HNTs@ZnγM Zero Charge Point (pHpzc)

An important parameter is the point of zero charge (pHpzc) of the adsorbents since it determines the pH at which the adsorbents surfaces have net electrical neutrality [16]. The technique of pH drift was applied to define the pHpzc of the Ch/g-HNTs@ZnγM tetra-nanocomposite, as shown in Figure 9. The results estimated that the PZC for the Ch/g-HNTs@ZnγM composite was pH of 10.9, indicating that the Ch/g-HNTs@ZnγM particles acquire a positive charge below this pH, while, above this pH, the particles acquire a negative charge.

#### 3.1.9. N_2_ Adsorption-Desorption

The nitrogen sorption measurements were performed to investigate the textural characteristics of resultant Ch/g-HNTs@ZnγM nanocomposites. The nitrogen isotherms in Figure 10 resulted in a type IV shape with an H_2_ hysteresis loop in the range of 0.3–0.98 relative pressure. These results suggest that the Ch/g-HNTs@ZnγM nanocomposites are characterized by mesoporous structures with pore sizes of 20–30 nm. These vicissitudes in hysteresis and pore size distribution may be ascribed to the role played by chitosan in tailoring the pore structure of the nanocomposites, which stems from integrating two-dimensional (chitosan sheets) and zero-dimensional (Zn γFe3O4 NPs) structures into a single material.

### 3.2. Effect of Parameters on Ions Removal

#### 3.2.1. Effect of Concentrations, at Constant Temperature = 30 °C and pH = 9

The effect of varying the mentioned ion concentrations (20 mg/L to 60 mg/L) was studied using the adsorbent Ch/g-HNTs@ZnγM while holding the other parameters such as pH = 9 and temperature = 30 °C. Figure 11 represents the concentrations of the ions versus the percentage of removal. This figure reveals that the Fe (III) removal % was increased by increasing the ion concentrations until 40 mg/L, corresponding to 99.06% removal efficiency then decreasing to 93.3% at 60 mg/L. The high ability of Ch/g-HNTs@ZnγM to adsorb Fe (III) may be due to that chitosan is a versatile polymer and has reactive (NH_2_ and-OH) groups on its backbone, which leads to a variety of applications and characteristics [42,43]. On the contrary, the adsorption of Cr (III) and Mn (II) by Ch/g-HNTs@ZnγM decreased by increasing the ion concentrations and reached the maximum removal of 95.2% and 87.1% at 20 mg/L. It is clear that the Ch/g-HNTs@ZnγM is slightly susceptible to the adsorption of Cr (III) and Mn (II) even with increasing their concentrations at fixed pH and temperature.

#### 3.2.2. Effect of pH at Constant Concentration = 20 mg/L and Temperature = 30 °C

One of the variables which influences the adsorption efficiency is solution pH since it defines the adsorbent surface charge, the ionization extent, and the adsorbent speciation [11]. The adsorption effect of Cr (III), Fe (III), and Mn (II) ions onto Ch/g-HNTs@ZnγM was studied in the pH range of 3.0–10.0 while maintaining the initial concentrations of the mentioned ions at 20 mg/L and the adsorbent Ch/g-HNTs@ZnγM dose at 2 g in 100 mL of solution at 30 °C. The effect of pH versus the percentage of the removed metal ions is shown in Figure 12. The results revealed that the removal percentage of the Cr (III) ion increased significantly with increasing the solution pH, as the highest removal percentage was 96% at the pH of 9, which could be related to the change in the surface charge distribution that significantly impacts metal ion removal. The adsorbent surface becomes positively charged at a low pH, pH = 2, because of H^+^ ion adsorption from the acidic medium, enhancing the adsorption of the negative loaded Cr (III) ions, which presents in the form of HCrO_4_^−^ at that pH. The iron (III) ions adsorption potential decreased with increasing the solution pH. As a result, increasing the adsorption ratio of metal ions was observed in the order of Cr (III) > Fe (III) > Mn (II).

#### 3.2.3. Effect of Temperatures at C_o_ = 20 mg/L and pH = 9

The operating temperatures at which heavy metal adsorption processes are carried out is a significant factor that influences adsorption ability and behaviour. Figure 13 shows the effect of adsorption temperature on Cr (III), Fe (III), and Mn (II) adsorption using Ch/g-HNTs@ZnγM as an adsorbent at a concentration of 20 mg/L, a pH of 9, and temperatures ranging from 300 to 350 K.

According to Figure 13 results, the adsorption removal rate decreases with the adsorption temperature increase, indicating the exothermic nature of the adsorption process, which agrees with the enthalpy change (ΔH) negative values. Consequently, at the temperature of 303.15 K (30 °C), the maximum removal percentages understudied occurred as 95.2%, 88.15%, and 87% for Cr (III), Fe (III), and Mn (II), respectively. The decrease in process efficiency could be attributed to the modifications and shrinkage of Ch/g-HNTs@ZnγM at higher temperatures, resulting in a lowering in the number of active sites. Furthermore, the adsorbed ions on the Ch/g-HNTs@ZnγM surface could detach them from the surface by increasing the temperature.

#### 3.2.4. Effect of Adsorbent Dosage (Conc. 20 mg/L and pH 9)

The amount of adsorbent dosage used in metal ion removal is crucial as the adsorbent’s equilibrium and the adsorbate are defined. The cost of adsorbent is also predicted. Figure 14 shows the experimental data regarding the adsorbent dosage effect on the % removal efficiency of Cr (III), Fe (III), and Mn (II) ions from their solution by Ch/g-HNTs@ZnγM while keeping all other parameters constant. It shows that with increasing the dose of adsorbent Ch/g-HNTs@ZnγM to 2 g, the removal efficiency % of Fe (III), Cr (III), and Mn (II) increased to be 96.9%, 97.5%, and 94.4%, respectively. The availability of more adsorbent surfaces for sorption may raise the metal ion removal rate at a high adsorbent dose. Previous studies have found similar behaviour caused by interactions between the metal ions and the adsorbent. The greater the region and the number of adsorption sites, the greater the overdose [42].

#### 3.2.5. Effect of Mixing Time

While holding all other parameters constant, the impact of mixing time on the adsorption removal of Cr (III), Fe (III), and Mn (II) ions onto Ch/g-HNTs@ZnγM was studied in the range of 20, 40, 60, and 100 min. Figure 15 shows that the studied metals’ removal efficiency was rapid between 20 and 60 min. By increasing time from 20 to 60 min, the removal efficiency % increases to 92%, 87%, and 94.8%, for Cr (III), Fe (III), and Mn (II) ions, respectively, with equilibrium being reached after 60 min for all the mentioned ions. The fast adsorption at the initial steps is regarded to the abundance of a large number of the surface-active sites on the adsorbent for the adsorption of the studied metals, which are used up over time and become saturated.

#### 3.2.6. Kinetics Study and Adsorption Modelling

This research included the two widely used kinetic models, pseudo-first-order and pseudo-second-order, to understand the adsorption process’s kinetic mechanism (rate and type) [43].

The pseudo-first-order kinetic model is represented by Equation (2) [44].

(2)
$$\ln(q_{e}-q_{t})=\ln q_{e}-k1.t/2.303$$


The pseudo-second-order model is given by Equation (3) [44].

(3)
$$q_{t}/t=\frac{1}{k_{2}q_{e}^{2}}+\frac{q_{e}}{t}$$

where, k_1_ is the pseudo-first-order rate constant (min^−1^) and *k*_2_ (g.mg^−1^.min^−1^) is the pseudo-second-order rate constant. *q_e_* (mg/g) and q_t_ refer, respectively, to the amount of metal ions adsorbed at equilibrium and at a time (*t*; min).

The linear curves of log (*q_e_*−*q_t_*) against time (min) and (*q_e_*/*t*) against time (min) were plotted to calculate the constant rate values (*k_1_*) and (*k_2_*), respectively, as presented in Appendix A. The two models’ corresponding parameters (*R^2^*, *K*, and *q_e_* (mg/g) of Cr (III), Fe (III), and Mn (II) ions adsorption on Ch/g-HNTs@ZnγM) are depicted in Table 3. The low *K_1_* value in the pseudo-first-order model indicates a slow adsorption rate, whereas the high *K_2_* value in the pseudo-second-order indicates an increase in the adsorption rates. The pseudo-second-order model regression coefficient (*R^2^* ≥ 0.96) had a higher *R^2^* value for the metal ion adsorption kinetics than the pseudo-first-order model. As a result, the pseudo-second-order model can be better fitted for the kinetics of Cr (III), Fe (III), and Mn (II) ions adsorption (Appendix A).

#### 3.2.7. Adsorption Isotherm Modeling

Two adsorption isotherm (Langmuir and Freundlich) models of Cr (III), Fe (III), and Mn (II) ions adsorption onto Ch/g-HNTs@ZnγM tetra-nanocomposite are displayed in Appendix A. The mentioned two models are used to evaluate the affinity of sorbent and adsorbate and explain the adsorption mechanism. The Langmuir and Freundlich models can be presented by Equations (4) and (5) [45].

(4)
$$\frac{C_{e}}{q_{e}}=\frac{1}{K_{L}q_{m}}+\frac{C_{e}}{q_{m}}$$


(5)
$$\;\log q_{e}=\log K_{f}+\left(1/n\right)\log C_{e}$$

where *q_e_* (mg/g) is the adsorption capacity at equilibrium, *C_e_* is the metal ion concentration at equilibrium (mg/L), and *q_m_* (mg/g) is the maximum monolayer adsorption capacity. *K_L_* (L/mg) is the Langmuir constant, *K_f_* (L/g) is the Freundlich constant, and *n* is the heterogeneity factor (index of the diversity, dimensionless).

The fundamental characteristic of the Langmuir isotherm model is represented in terms of separation factor (*R_L_*), a dimensionless equilibrium parameter, described by Equation (6) [46].

(6)
$$R_{L}=1/\left(1+C_{o}\times K_{l}\right)$$


*C_o_* (mg/L) is the amount of initial adsorbate and *K_L_* (L/mg) is the separation factor. The parameter *R_L_* is considered to be a more accurate adsorption indicator. The value of *R_L_* suggested whether the adsorption is irreversible (*R_L_* = 0), favorable (0 < *R_L_* < 1), linear (*R_L_* = 1), or unfavorable (*R_L_* > 1).

By drawing the relation between log *C_e_* and log *q_e_* displayed in Appendix A, the Freundlich and exponent constants (*K_f_* and *n*) are determined from the slope and intercept and of the straight lines. The estimated isotherm model constants and the related coefficients of correlation values are summarized in Table 4. Compared to the Langmuir isotherm model, the correlation coefficient “R^2^” of the Freundlich isotherm was far from unity. The equilibrium data for Cr (III), Fe (III), and Mn (II) adsorption on Ch/g-HNTs@ZnγM have fitted the Langmuir model better than the Freundlich model, as shown in Appendix A. Langmuir model suggests that monolayer adsorption occurs due to a uniform distribution of active sites around the adsorbent surface.

#### 3.2.8. Thermodynamic Adsorption Parameters

The thermodynamic parameters for adsorption and the values of *K_eq_* (Langmuir constant; L/mg) at different temperatures (*T*; *K*) were processed according to the Van ’t Hoff equation (Equation (7)) [47].

(7)
$$\ln K_{eq}=\frac{\Delta S^{o}}{R}+\frac{\Delta H^{o}}{RT}$$

where Δ*H*° (J/mol) and Δ*S*° (J/mol.K) are enthalpy and entropy changes, respectively.

Plotting ln *K_eq_* against 1/*T* gives straight lines with slopes and intercepts equal to Δ*H*°/*R* and Δ*S*°/*R*, respectively, from which enthalpy and entropy changes can be calculated as shown in Appendix A.

The Gibbs free (∆*G*°; J/mol) of adsorption was calculated from the following relation [48] (Equation (8)):
(8)
$$\Delta G^{o}=\Delta H^{o}-T\Delta S^{o}$$


Table 5 shows the thermodynamic parameters of the reaction. For all metal ions, the entropy change is positive, indicating that the reaction is less random. In contrast, the enthalpy change and the ∆*G*° values are negative, indicating, respectively, that the process is exothermic and spontaneous with a favourable sorption process.

## 4. Conclusions

In this paper, a novel Ch/g-HNTs@ZnγM magnetic quaternary nanocomposite was fabricated to remove the ions of Cr (III), Fe (III), and Mn (II)) from wastewater. Its characteristics were investigated utilizing FTIR, SEM, XRD, GPC, TGA, TEM, and surface zeta potential. Adsorption investigations were carried out at various conditions to determine their effects on the adsorption process and obtain the isotherms of reaction. The maximum removal % of Cr (III), Fe (III), and Mn (II) on the Ch/g-HNTs@ZnγM adsorbent reached 95.2%, 99.06%, and 87.1% at 40 mg/L of ion concentrations. The optimum removal for all studied metal ions was obtained at a pH of 9.0 and a contact time of 60 min, which were utilized for equilibrium removal. The thermodynamic study of adsorption was with negative values of ΔH°, ΔG°, and positive value of ΔS° indicates an exothermic, spontaneous, and chemical adsorption. For isotherm and kinetic modelling, the Langmuir isotherm and pseudo-second-order models were better fitted the experimental data.

Additionally, upon completion of the adsorption process in contaminated water, the magnetic properties of the synthesized tetra-nanocomposite represent an advantage for recovering its particles. Thus, the current study suggests the Ch/g-HNTs@ZnM tetra- nanocomposite as a highly efficient, promising, and green alternative source adsorbent in the wastewater treatment concept. Furthermore, by using Ch/g-HNTs@ZnγM for Cr (III), Fe (III), and Mn (II) adsorption, the environmental load, and effect of Ch/g-HNTs@ZnγM will be reduced, and so the impact of Ch/g-HNTs@ZnγM will via an environmentally friendly method.

## Figures and Tables

**Figure 1 polymers-13-02714-f001:**
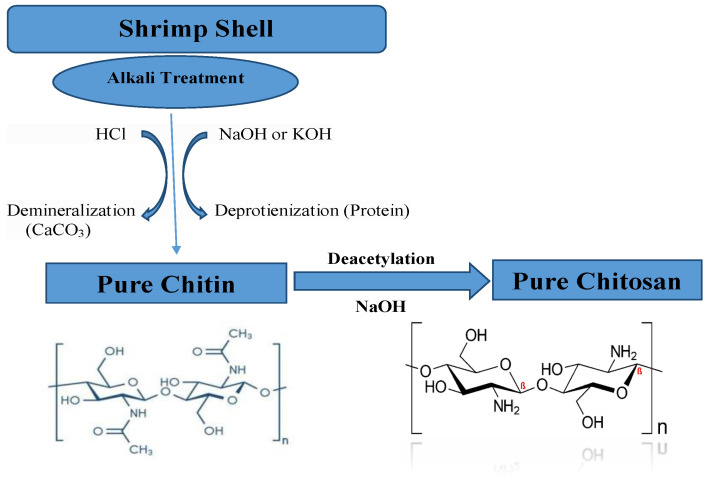
Simplified chemical extraction method of chitin and chitosan from shrimp shells.

**Figure 2 polymers-13-02714-f002:**
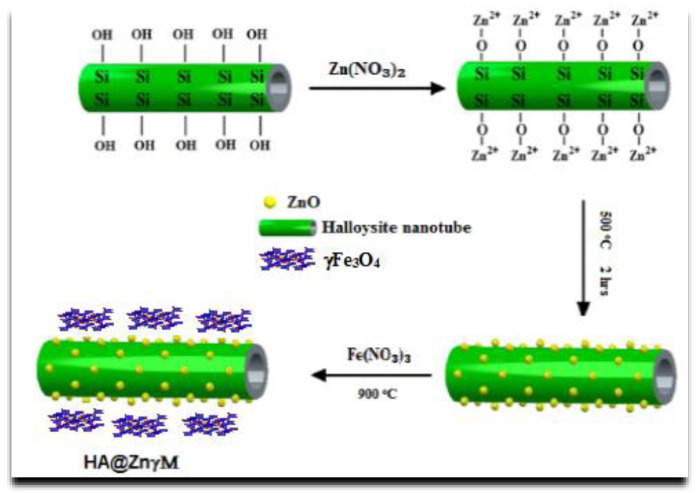
Scheme of the synthesis of halloysite/ZnγFe_3_O_4_ core/shell (HNTS/ZnγM).

**Figure 3 polymers-13-02714-f003:**
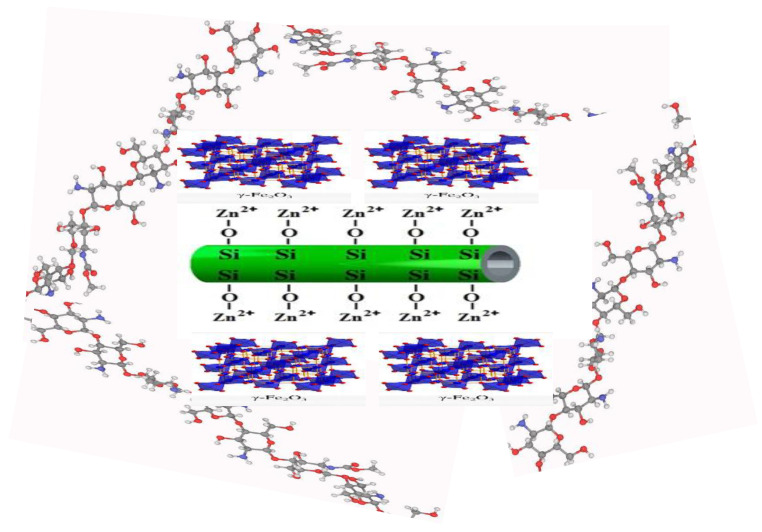
Scheme of the predicted shape of the synthesized quaternary magnetic nanocomposite Ch/g-HNTs@ZnγM.

**Figure 4 polymers-13-02714-f004:**
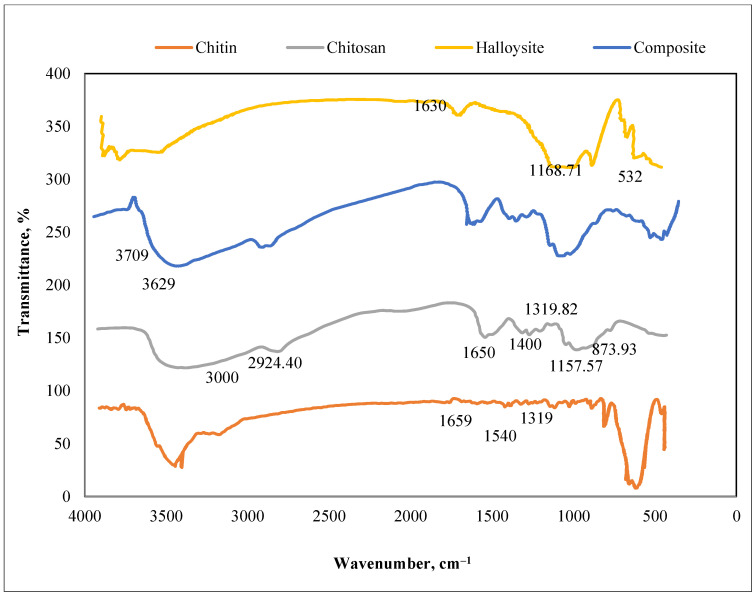
FTIR spectra of chitin, chitosan, halloysite, and composite.

**Figure 5 polymers-13-02714-f005:**
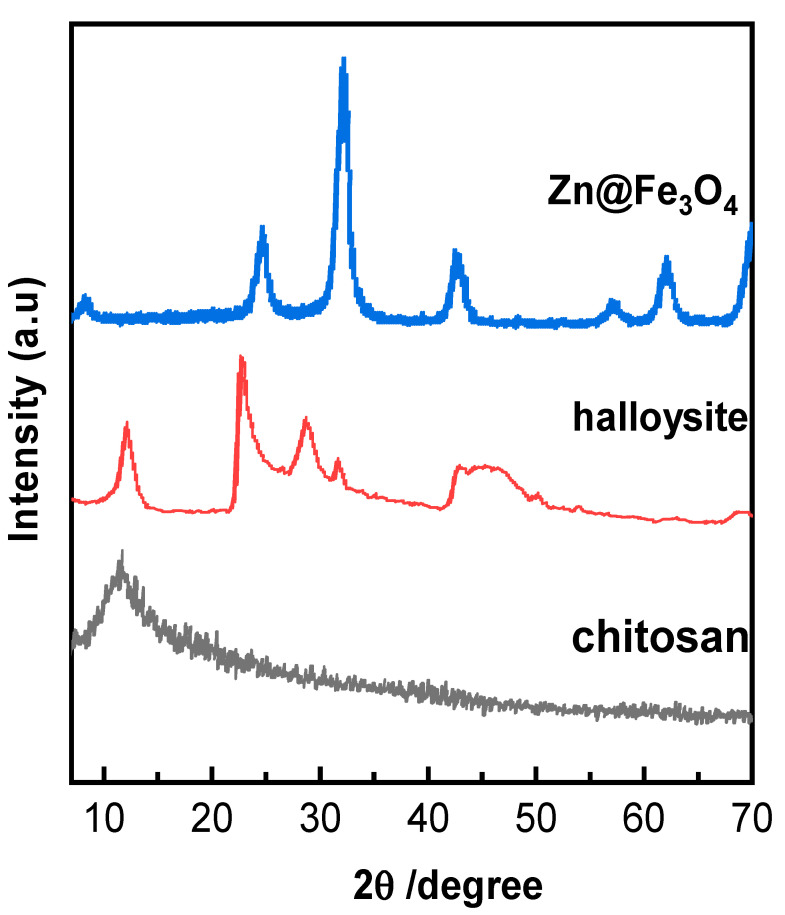
X-ray diffraction (XRD) patterns for chitosan, halloysite, Zn@Fe_3_O_4_, and composite.

**Figure 6 polymers-13-02714-f006:**
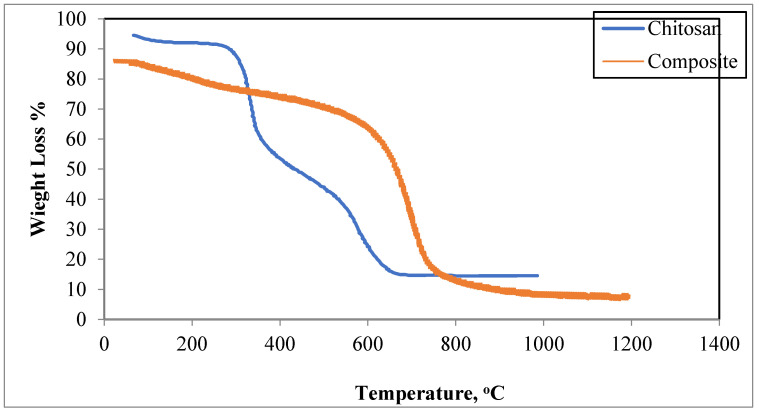
Thermogravimetric analysis (TGA) for chitosan and composite.

**Figure 7 polymers-13-02714-f007:**
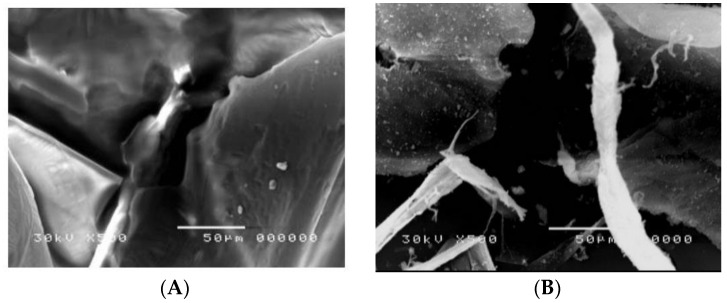
SEM micrographs for (**A**) chitin, (**B**) chitosan, and (**C**) Ch/g-HNTs@ZnγM composite.

**Figure 8 polymers-13-02714-f008:**
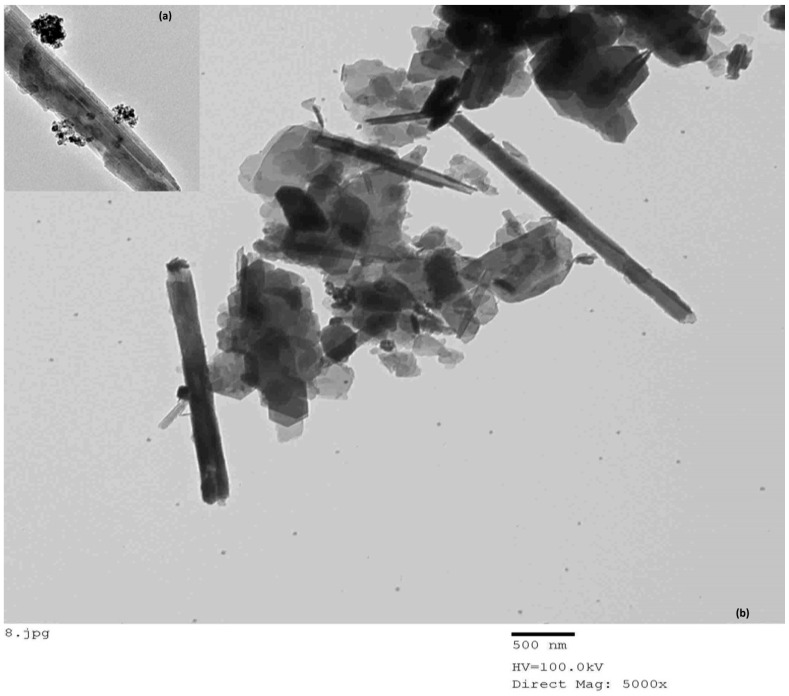
TEM of (**a**) halloysite and (**b**) magnetic halloysite.

**Figure 9 polymers-13-02714-f009:**
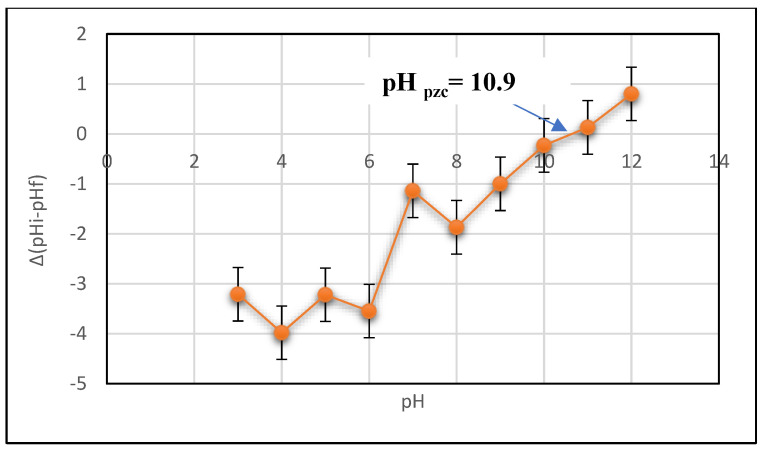
Point of zero charge (pH_PZC_) of the Ch/g-HNTs@ZnγM, determined by the pH drift technique.

**Figure 10 polymers-13-02714-f010:**
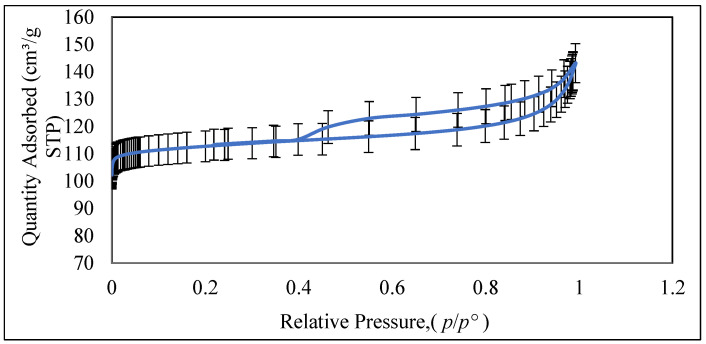
Nitrogen adsorption–desorption isotherms of Ch/g-HNTs@ZnγM nanocomposites.

**Figure 11 polymers-13-02714-f011:**
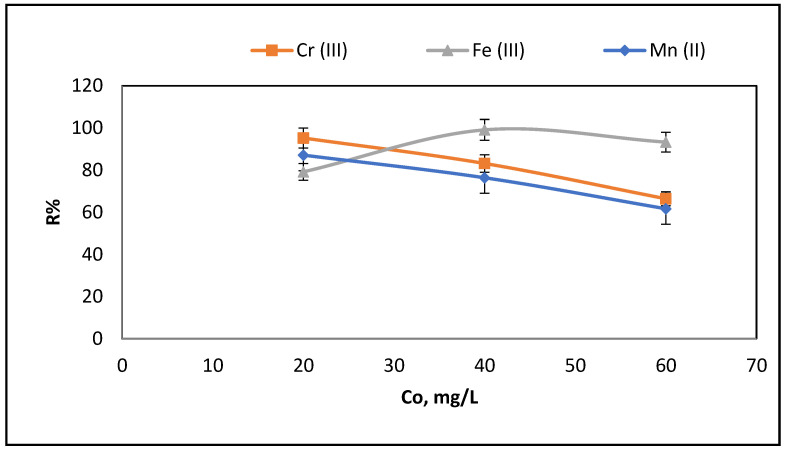
Effect of initial metals concentrations on the Cr (III), Fe (III) and Mn (II) removal onto Ch/g-HNTs@ZnγM at 30 °C and pH 9.

**Figure 12 polymers-13-02714-f012:**
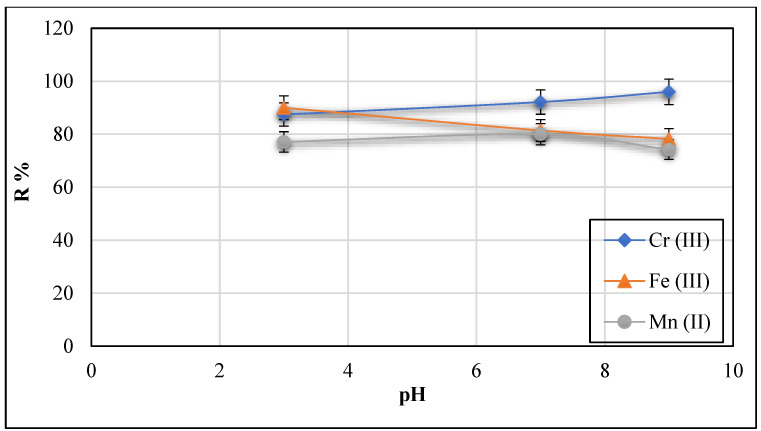
Effect of solution pH on Cr (III), Fe (III), and Mn (II) adsorption onto Ch/g-HNTs@ZnγM adsorbent at 30 °C and concentration of 20 mg/L.

**Figure 13 polymers-13-02714-f013:**
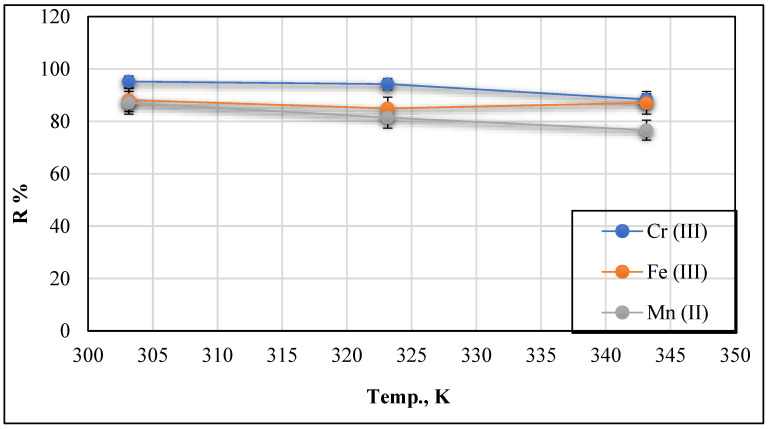
Effect of temperature on Cr (III), Fe (III), and Mn (II) removal at Conc. 20 mg/L and pH 9.

**Figure 14 polymers-13-02714-f014:**
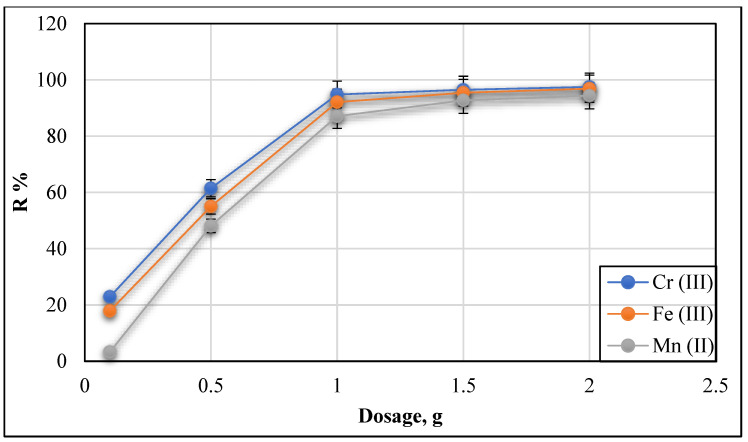
Adsorbent Ch/g-HNTs@ZnγM dosage versus removal % of Cr (III), Fe (III), and Mn (II).

**Figure 15 polymers-13-02714-f015:**
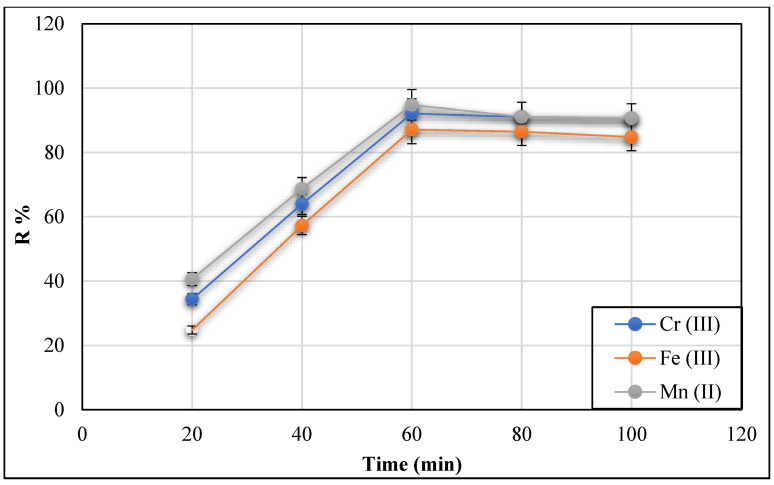
Effect of mixing time on Cr (III), Fe (III), and Mn (II) removal onto Ch/g-HNTs@ZnγM adsorbent.

**Table 1 polymers-13-02714-t001:** GPC for chitosan sample after 60 min.

Retention Time	M_n_	M_w_	M_p_	M_z_	M_z+1_	Polydispersity
27.220	50,015	86,380	50,493	144,904	208,744	1.727065

**Table 2 polymers-13-02714-t002:** GPC for chitosan sample after 120 min.

Retention Time	M_n_	M_w_	M_p_	M_z_	M_z+1_	Polydispersity
26.112	51,518	110,534	90,177	193,865	260,533	2.145541

**Table 3 polymers-13-02714-t003:** Constants of the pseudo-first-order and pseudo-second-order models.

Kinetic Order	Parameters	Metal Ions		
		Cr (III)	Fe (III)	Mn (II)
Pseudo-first-order	*q_e_* (mg/g)	1.0234	1.0268	1.01547
	R^2^	0.85433	0.29425	0.89897
	*K_1_* (min^−1^)	0.01478	0.0019	0.01023
Pseudo-second-order	*q_e_* (mg/g)	2.4078	2.40781	2.63299
	R^2^	0.98978	0.96695	0.9878
	*K_2_* (min^−1^)	1.626199	1.23978	0.70769

**Table 4 polymers-13-02714-t004:** Adsorption isotherms constants for adsorption of Cr (III), Fe (III), and Mn (II) on (Ch/g-HNTs@ZnγM).

Model	Constants	Metal Ions		
		Cr (III)	Fe (III)	Mn (II)
Langmuir	*q_m_* (mg/g)	0.893395	0.4467	0.23182
	*K_L_* (L/mg)	0.231614	0.115807	0.107141
	R^2^	0.999989	0.999989	0.999700
Freundlich				
	*K_f_*	0.753768	0.351077	0.283438
	*n*	0.408312	0.41059	0.57226
	R^2^	0.99261	0.98584	0.994095

**Table 5 polymers-13-02714-t005:** Thermodynamic constants for the adsorption of Cr (III), Fe (III), and Mn (II) on the Ch/g-HNTs@ZnγM at various temperatures.

	**Cr (III)**		
*T* (*K*)	Δ*H* (J/mol)	Δ*S* (J/mol.*K*)	Δ*G* (J/mol)
303.15	−35,703.1	110.688	−69,258.1
323.15	−35,703.1	110.688	−71,471.9
343.15	−35,703.1	110.688	−73,685.6
	**Fe (III)**		
*T* (*K*)	Δ*H* (J/mol)	Δ*S* (J/mol.*K*)	Δ*G* (J/mol)
303.15	−30,960.4	99.57985	−61,148
323.15	−30,960.4	99.57985	−63,139.6
343.15	−30,960.4	99.57985	−65,131.2
	**Mn (II)**		
*T* (*K*)	Δ*H* (J/mol)	Δ*S* (J/mol.*K*)	Δ*G* (J/mol)
303.15	−43,996.5	135.2966	−85,011.6
323.15	−43,996.5	135.2966	−87,717.6
343.15	−43,996.5	135.2966	−90,423.5

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
