# Peer review of "Enhanced Performance of Chitosan via a Novel Quaternary Magnetic Nanocomposite Chitosan/Grafted Halloysitenanotubes@ZnγFe3O4 for Uptake of Cr (III), Fe (III), and Mn (II) from Wastewater"

_polymers, 2021, doi:10.3390/polym13162714_

Round 1

Reviewer 1 Report

This manuscript demonstrates the enhanced performance of chitosan via a novel quaternary magnetic nanocomposite chitosan/grafted halloysitenanotubes@ZnγFe3O4 for uptake of Cr (III), Fe (III), and Mn (II) from wastewater. A major revision is needed because of the following reasons.

  1. ZnyFe3O4 presented in the title and other places in the manuscript should be explained because it is so weird.
  2. The introduction should be improved in terms of presenting more advantages of chitosan for metal removal when comparing other adsorbents. Here are some references that may be useful: Environmental Science & Technology 2021 55 (8), 4287-4304; ACS ES&T Engineering 2021 1 (4), 623-661; Journal of Cleaner Production 2019, 232, 774-783.
  3. Higher quality Figures are needed.
  4. Figure 4-FIIR spectra should contain information on peak assignment.
  5. N2 adsorption-desorption cures should be measured to analyze the adsorbent's porosity.
  6. Modeling of Cr(III) in Figure 15 should be checked because the point of "0" at 60 min is very unusual.
  7. Some figures should be removed to supporting information.
  8. Cr (III) and other similar typos should be revised into right one "Cr(III)", please check through the whole manuscript carefully.
  9. English must be improved.  

Author Response

Dear reviewer; thanks for your comments.

All comments have been answered.

Reviewer 2 Report

The present manuscript presents the use of new sorbents in order to remove the metal ions from wastewaters.

This manuscript can be accepted after a major revision. All the manuscript have to be corrected from orthographic point of view. There are to many mistakes.

1. English language need to be improved.

2. Correct is pH not PH, mmol not mmoL, K (for temperature not oK).

3. Scheme 3 is very unclear.

4. I consider that a dose of 2g/100 mL solution is inefficient from economic pint of view.

5. Revise title of Fig. 13 from orthographic point of view.

6. Figs. 15, 16 must to be added the measurement units for time.

7. Why the qm value for Fe(III) is negative? That could means that Fe(III) is not removed from water?

8. Eq (7) is wrong.

9. Why the authors choose pH 9 for the sorption? Because the maximum sorption for Fe(III) and Mn(II) is not higher at this value.

10. What means GPC analysis? The authors need to explain the all the abbreviations.

11. Why they choose the initial concentration of ions 20 mg/L?

12. In order to evidence the efficiency of the new sorbents the authors have to compare their results with the results from literature. I suppose that their obtained values are not so high.

13. Can be regenerated these sorbents?

14.What is the measurement unit for enthalpy, J/mol or kJ/mol? (line 437, Table 5)

15. The References section should be revised all. Some journal have entire name and some not. Some titles of the journals are improper written. (For example correct is: International Journal of Biological Macromolecules).

Author Response

Thanks for these comments.

All comments have been answered.

Round 2

Reviewer 1 Report

This manuscript is highly improved. I have no comments. It's ready for publication.

Author Response

All the comments have been done.
